# 🦉*MFC-Bench*: BENCHMARKING MULTIMODAL FACT-CHECKING WITH LARGE VISION-LANGUAGE MODELS

**Shengkang Wang**[1,*]**, Hongzhan Lin**[2,*]**, Ziyang Luo**[2,*]**, Zhen Ye**[3]**, Guang Chen**[1,†]**, Jing Ma**[2,†]

[1]Beijing University of Posts and Telecommunications
[2]Hong Kong Baptist University
[3]Hong Kong University of Science and Technology
{wsk, chenguang}@bupt.edu.cn, {cshzlin, cszyluo, majing}@comp.hkbu.edu.hk

## ABSTRACT

Large vision-language models (LVLMs) have significantly improved multimodal reasoning tasks, such as visual question answering and image captioning. These models embed multimodal facts within their parameters, rather than relying on external knowledge bases to store factual information explicitly. However, the content discerned by LVLMs may deviate from factuality due to inherent bias or incorrect inference. In this work, we introduce *MFC-Bench*, a rigorous and comprehensive benchmark designed to evaluate the factual accuracy of LVLMs across three stages of verdict prediction for multimodal fact-checking (MFC): Manipulation, Out-of-Context, and Veracity Classification. Through our evaluation on *MFC-Bench*, we benchmarked a dozen diverse and representative LVLMs, uncovering that current models still fall short in MFC and demonstrate insensitivity to various forms of manipulated content. We hope that *MFC-Bench* could raise attention to the trustworthy AI potentially assisted by LVLMs in the future.

## 1 INTRODUCTION

Recent advancements in natural language processing (NLP), particularly with large language models (LLMs) (Chang et al., 2023), have introduced tools like ChatGPT and GPT-4 (OpenAI, 2023) that excel in understanding human instructions using strategies such as instruction tuning and reinforcement learning from human feedback (Ouyang et al., 2022). These models demonstrate strong zero-shot or few-shot capabilities, performing tasks without additional fine-tuning (Kojima et al., 2022; Lin et al., 2023). Simultaneously, large vision-language models (LVLMs) (Dai et al., 2023; Gong et al., 2023) have extended this proficiency to multimodal understanding tasks (Fu et al., 2023). These advancements mark a significant step forward in artificial intelligence, enabling more cohesive applications across modalities.

Recent studies have thoroughly investigated the extent to which LLMs hold factual information and their capacity to reason with such knowledge (Hu et al., 2024), which hypothesized that LLMs, trained on vast data, could adequately substitute for evidence retrieval and conduct fact-checking autonomously, relying solely on their parametric knowledge. Beyond text-only fact-checking (Thorne et al., 2018; Lin et al., 2022a; Guo et al., 2022), multimodal content is often perceived as more credible and spreads more quickly than similar textual claims (Li & Xie, 2020; Newman et al., 2012). However, the capabilities and limitations of LVLMs in managing multimodal reasoning tasks (Akhtar et al., 2023) related to factuality, particularly in identifying online unverified information within multimodal inputs, remain underexplored. These multimodal fact-checking tasks (Nakamura et al., 2020; Shao et al., 2023; Yao et al., 2023) are crucial for understanding social dynamics and require sophisticated social judgment and decision-making abilities. Thus, a fundamental question remains: *Can LVLMs discern factuality in a multimodal context?* Given that LVLMs are trained on extensive and varied image-text corpora and demonstrate remarkable generalization capabilities (Liu et al., 2023a), it is vital to evaluate both their strengths and potential challenges

---

[*]Equal contribution.
[†]Corresponding authors.

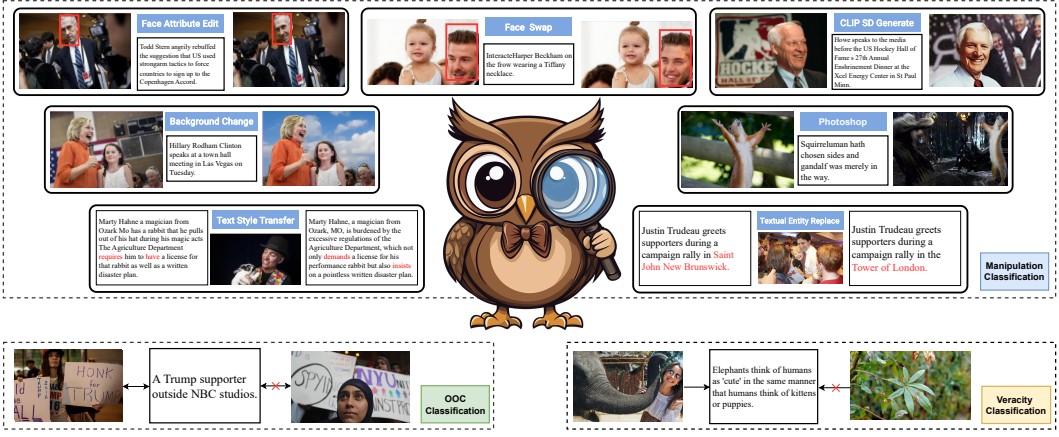

Figure 1: *MFC-Bench* is a comprehensive benchmark designed to evaluate the LVLMs across three stages of verdict prediction for MFC: Manipulation Classification, Out-of-Context Classification, and Veracity Classification.

in handling factual knowledge and reasoning. This inquiry is particularly relevant to ensuring trustworthy insights, focusing on how LVLMs analyze and integrate complex visual and textual elements accurately and responsibly.

Previous literature (Akhtar et al., 2023) has surveyed that there are three important stages for verdict prediction in multimodal fact-checking process: 1) Manipulation Classification; 2) Out-of-Context (OOC) Classification; 3) Veracity Classification. In this work, we aim to comprehensively explore the helpfulness of **LVLMs** in benchmarking multimodal fact-checking within these three tasks. To this end, we introduce *MFC-Bench* as shown in Figure 1, a comprehensive Multimodal Fact-Checking testbed designed to evaluate LVLM in terms of identifying factual inconsistencies and counterfactual scenarios. *MFC-Bench* encompasses a wide range of visual and textual queries, organized into the three verdict prediction tasks: Manipulation Classification, OOC Classification, and Veracity Classification. These three sub-tasks of multimodal fact-checking draw from a mix of diverse datasets (Newman et al., 2012; Shao et al., 2023; Yao et al., 2023) and our newly created datasets specifically designed for analyzing awareness of multimodal facts: 1) The Manipulation Classification task targets various alterations like face swapping, face attribute editing, background changing, image generation, photoshop, entity replacement, and style transfer; 2) The OOC Classification task focuses on identifying the false connection between the image and text that may be both true; 3) The Veracity Classification task is the multimodal counterpart to classifying the veracity of *textual claims* given the visual evidence, by leveraging the inherent knowledge embedded in LVLMs. As a result, such a three-task design philosophy could facilitate evaluating the effectiveness of LVLMs in supporting verdict prediction during the multimodal fact-checking process. We presented MFC tasks to LVLMs with carefully crafted prompts, gathered the model's feedback, and conducted a comprehensive analysis of the outcomes, which ensures a thorough understanding of LVLMs' capabilities and limitations on *MFC-Bench*.

Through *MFC-Bench*, we comprehensively assess the ability of various LVLMs (Bai et al., 2023; Dai et al., 2023; Liu et al., 2023a; OpenAI, 2023) to accurately identify manipulated and misleading content within multimodal inputs. Our benchmark offers a rigorous examination of current LVLMs, highlighting the considerable gaps in their performance. Tasks aimed at detecting false connections, such as OOC Classification, reveal pronounced disparities in LVLM efficacy. For more intricate tasks like Manipulation Classification, which necessitates deep background knowledge and sophisticated reasoning, LVLMs typically demonstrate only mediocre performance. Besides, we further explore the justification production of LVLMs for multimodal fact-checking with human subject evaluation. Overall, *MFC-Bench* is designed to provide researchers with a multi-dimensional understanding of their LVLMs' capabilities in multimodal fact-checking. Our goal is to advance auditing insights within LVLMs, playing a crucial role in curbing the spread of online disinformation and promoting the stability and cohesion of diverse communities.

Table 1: Dataset sources, description, and distribution.

| Types | Description | Sources | Distribution | | |
|-------|-------------|---------|------|-----------|-----|
| | | | Fact. | Non-Fact. | All |
| **Manipulation** | Face Swap | DGM4 (Shao et al., 2023) | 4,000 | 2,000 | 6,000 |
| | Face Attribute Edit | DGM4 (Shao et al., 2023) | 4,000 | 2,000 | 6,000 |
| | Background Change | - | 1,000 | 2,000 | 3,000 |
| | CLIP-based SD Generate | - | 5,000 | 5,000 | 10,000 |
| | Photoshop | Fakeddit (Nakamura et al., 2020) | 1,000 | 1,000 | 2,000 |
| | Textual Entity Replace | - | 1,162 | 838 | 2,000 |
| | Text Style Transfer | - | 1,000 | 1,000 | 2,000 |
| **OOC** | Detect out of context | NewsCLIPpings (Luo et al., 2021) | 1,000 | 1,000 | 2,000 |
| **Veracity** | Verify the claim w/ image | Mocheg (Yao et al., 2023) | 469 | 1,531 | 2,000 |

Our contributions are three-fold: 1) We introduce *MFC-Bench*, a comprehensive testbed with 35K multimodal samples across three stage sub-tasks of verdict prediction in the multimodal fact-checking process to assess LVLMs' trustworthiness; 2) Extensive evaluation of a dozen advanced LVLMs reveals significant challenges, with GPT-4o only achieving F1 scores of 69.4% on the *MFC-Bench*; 3) We provide a detailed analysis of performance variations among different LVLMs on prompting strategies and justification production.

## 2 DATASET CONSTITUTION

To systematically assess the visual and textual factual knowledge related to inconsistencies and counterfactual reasoning abilities of LVLMs, we have formulated our benchmark into three decomposed sub-tasks of verdict prediction for the multimodal fact-checking process: Manipulation Classification, Out-of-Context Classification, and Veracity Classification, by considering prevalent multimodal misinformation types (Akhtar et al., 2023).

For these multimodal misinformation types of data for verification, we carefully curate appropriate visual and textual queries from a variety of sources to ensure a comprehensive evaluation of LVLMs in multimodal fact-checking, as summarized in Table 1.

### 2.1 MFC DATA TYPES

#### 2.1.1 MANIPULATION CLASSIFICATION

Manipulation Classification is a task meticulously designed to ascertain whether multimodal data encompasses fabricated elements (Qi et al., 2019) by using LVLMs. To investigate LVLMs' proficiency in identifying multimodal content altered through various manipulative techniques, in *MFC-Bench*, we utilized seven types of manipulation methods[1]: The first five focus on visual alterations, while the last two target textual modifications.

**Method 1: F̲a̲ce S̲wap (FS).** Face Swap involves the process of cutting a face from one image and replacing it with a different face in another image. As shown in Figure 1, through the use of face swap, Beckham's face has been replaced with a different face. We include the Face Swap data to *assess whether LVLMs can recognize public figures and retrieve information related to individuals, finding the counterfactuals that emerge from these swapped faces* in the multimodal context.

**Method 2: Face A̲ttribute E̲dit (AE).** Face Attribute Edit achieves deception by altering the facial expressions of humans like newsmakers. For example, in Figure 1, Todd Stern originally had an angry expression, which was changed to a happy expression through Face Attribute Edit. This inclusion allows us to *evaluate the multimodal fact-checking capabilities of LVLMs in recognizing the scene, identifying personal information and detecting the correctness of face's status* in visual content assisted with an accompanying text.

---

[1]Here, we consider the most challenging setting (Akhtar et al., 2023) that the correct content in one modality, accompanied by the manipulated content in the other modality, which increases credibility.

**Method 3: Background Change (BC).** Background Change alters images, transforming public individuals into scenes where he/she never showed up in reality. As depicted in Figure 1, Hillary Rodham Clinton was originally indoors, but BC makes it seem like she is now outside. The objective is to *examine the capability of LVLMs for accurate identification of individuals and scenes in images, evaluating their correspondence and authenticity in relation to the descriptions provided in texts.*

**Method 4: CLIP-based Stable Diffusion Generate (CG).** CLIP-based Stable Diffusion (Ramesh et al., 2022) features an image-to-image generation pipeline that enables the manipulated image to retain the linguistic information from the original image, producing stable-diffusion versions for image replacement. Originally, Figure 1 showed Howe speaking, but with the CG method, the image was altered to display a generated individual giving the speech, retaining much of the original visual content. This design enables us to *assess the fact-checking capabilities of LVLMs regarding their awareness of whether multimodal content is fabricated, even when the manipulated image retains elements of the original alongside the raw text.*

**Method 5: Photoshop (PS).** Photoshop has long been a leading manipulation for manual image editing, enabling users to alter human figures and merge different images to create potentially misleading visuals. As demonstrated in Figure 1, using Photoshop, an ordinary squirrel can be seen battling Gandalf in a single picture. Including this data type allows us to *assess whether LVLMs can discern the traces of human manipulation in image accompanying the original text.*

**Method 6: Textual Entity Replace (ER).** Textual Entity Replace involves substituting entities other than the target persons in the data, with randomly chosen locations and time. As exemplified in Figure 1, Justin Trudeau was originally shown greeting in Saint John, New Brunswick, Canada, but with Textual Entity Replace, it was changed to him greeting at another country, the Tower of London, British. This method seeks to *assess the capability of LVLMs to effectively associate individuals with the entities depicted in both images and texts, discerning any inconsistencies with multimodal facts in commonsense.*

**Method 7: Text Style Transfer (ST).** Text Style Transfer is the process of modifying the tone and style of a text to alter the perception of the same person or event, potentially leading to a different factual impression (Wu et al., 2024). As Figure 1, by Text Style Transfer, the tone shifts from a neutral, factual statement about Marty Hahne needing a license and disaster plan for his rabbit, to a more critical and dramatic tone, portraying the requirements as burdensome and excessive. The process *examines LVLMs' ability to rigorously comprehend the events and associated sentiments depicted in images and claims, and to correctly correlate them.*

### 2.1.2 Out-of-Context Classification

Out-of-Context (OOC) Classification in *MFC-Bench* aims to decipher the coherence and correspondence of context across various modalities (Luo et al., 2021) with LVLMs. We collected multimodal samples from the NewsCLIPpings dataset (Luo et al., 2021). Unlike the aforementioned manipulation techniques that require modifying images and texts, OOC Classification combines real but misused images and texts. If the image and the text are contextually aligned, the relationship is regarded as true, naturally representing fact. Conversely, if the image and the text are not contextually aligned, the relationship is regarded as false, indicating non-fact.

### 2.1.3 Veracity Classification

Veracity Classification in *MFC-Bench* serves to classify the factuality of textual claims based on visual evidence (Yao et al., 2023) by employing LVLMs. Based on the image evidence, the LVLMs need to predict the truthfulness of the textual claim. We curated a subset of the Mocheg dataset (Yao et al., 2023) for this task. If the image evidence supports the truthfulness of the textual claim, the relationship between the image and the claim is supported, indicating fact. Otherwise, the claim is treated as refuted by the image, exhibiting non-fact.

### 2.2 Label Setting

To unify the three tasks and facilitate a more effective analysis of benchmark results, we formulate the tasks into binary classification, we define the label $L = \{\texttt{Fact.}, \texttt{Non-Fact.}\}$. The Manipulation Classification task involves determining whether multimodal news is fabricated, with labels indi-

cating "`Manipulated`" (Non-Fact.) or "`Not Manipulated`" (Fact.). The OOC Classification task assesses whether the image and claim are inconsistent, with labels indicating "`Matched`" (Fact.) or "`Not Matched`" (Non-Fact.). The Veracity Classification task evaluates whether the claim is true based on image evidence, with labels indicating "`Supported`" (Fact.) or "`Refuted`" (Non-Fact.).

## 2.3 QUALITY ASSURANCE

Multiple levels of measures are implemented to guarantee data reliability. First, we utilize established and reputable technologies such as Stable Diffusion and GPT-4 for data processing, ensuring that the operations are reasonable and aligned with our expectations. Second, we incorporate other well-regarded datasets that are time-tested and frequently cited. The tasks represented by these datasets coincide with the objectives of our benchmark. Third, after constructing the dataset, we conduct a Human Quality Check by performing partial sampling. Specifically, we randomly select 100 entries from each new category (i.e., BC, CG, ER, and ST) to verify the self-constructed dataset's integrity and ensure the effectiveness of the manipulation methods we have applied. Finally, our benchmarking includes two types of human-involved experiments. The first type involves comparing the LVLM's performance to human performance; the second type entails human subject evaluation of the LVLM's performance based on its justification production.

**Human Quality Check** This research involved a human subject study to evaluate the quality of multimodal data manipulated by our adopted techniques. To assure the quality of the self-constructed data, we employed three human evaluators, who are senior undergraduate or graduate students majoring in computer science. Each evaluator was presented with the manipulated data and the original data to judge whether the data had been successfully manipulated using manipulation techniques for the reliability and credibility of the multimodal data. Each evaluator completed the quality assurance process independently. Further details regarding the evaluation process are provided in Appendix §C.2. The manipulation accuracy for each task is presented in Table 2, which highlights the effectiveness of our techniques. Additionally, the intra-class agreement score is 0.705. The average Spearman's correlation coefficient between any two annotators is 0.714. These figures reflect the reliability of our data manipulation methods and the consistency of the evaluators' assessments.

Table 2: Manipulation Accurary for Different Types.

| Types | Accuracy |
|---|---|
| Background Change | 0.97 |
| CLIP-based SD Generate | 1.00 |
| Textual Entity Replace | 0.99 |
| Text Style Transfer | 0.98 |

## 3 METHODOLOGY

### 3.1 MODELS

To provide an exhaustive perspective on the current state of emerging LVLMs within the context of multimodal fact-checking, we conducted comprehensive evaluations on representative accessible LVLMs. Our selection encompasses a range of models from diverse organizations, differing in size, which allows for a thorough understanding of the capabilities and limitations of LVLMs in handling multimodal content concerned with actuality For the open-source and accessible LVLMs, we adopt the representative models like Emu2 (Sun et al., 2023), InternVL (Chen et al., 2023c), CogVLM (Wang et al., 2023a), LLaVA-NeXT (Liu et al., 2024a), InstructBLIP (Dai et al., 2023), Pixtral[2], MiniCPM-V-2.6 (Yao et al., 2024), LLaVA-OneVision (Li et al., 2024a), Molmo (Deitke et al., 2024), Qwen-VL (Bai et al., 2023), Qwen2-VL (Wang et al., 2024b), Yi-VL (Young et al., 2024) and xGen-MM (Xue et al., 2024). As five of the most powerful closed-source LVLMs, GPT-4o, GPT-4V, Claude3.5-Sonnet, Claude3-Haiku and Gemini-1.5-Pro are included in our testing scope.

---

[2]https://mistral.ai/news/pixtral-12b/

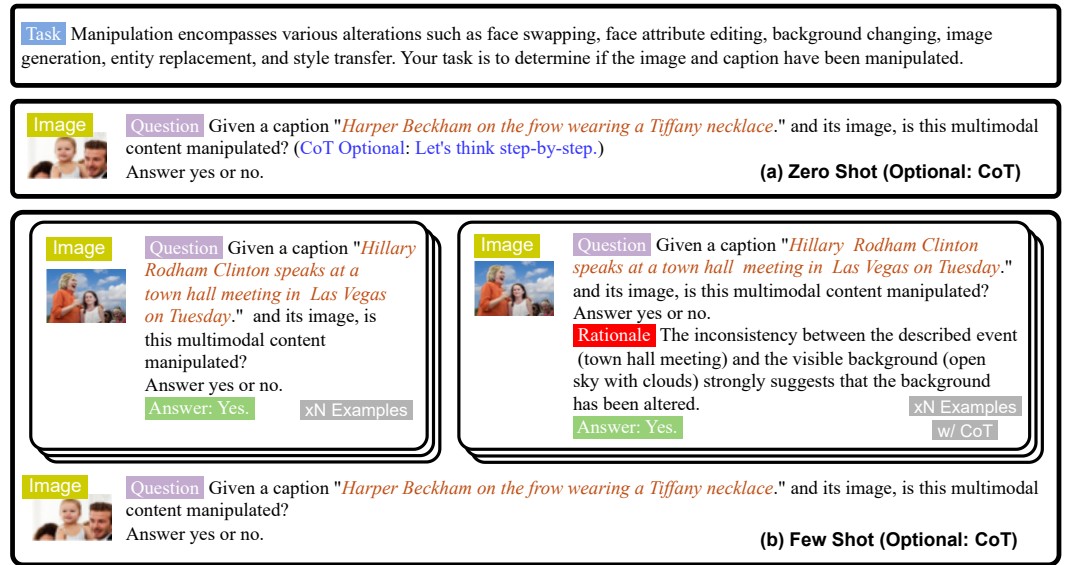

Figure 2: Comparison of prompts in zero-shot and few-shot scenarios with and without CoT.

## 3.2 PROMPT STRATEGY

We define a multimodal content $M = \{I, C\}$ as a tuple consisting of an image $I$ and an accompanying textual claim $C$ to be fact-checked.

Given that our benchmark comprises three important decomposed sub-tasks for verdict prediction in the MFC process (Akhtar et al., 2023), we have developed three task instructions $T_i$ specifically designed to elicit the multimodal fact-checking capabilities of the LVLMs as follows:

**Manipulation Classification (Task $T_1$):** "*Manipulation encompasses various alterations such as face swapping, face attribute editing, background changing, image generation, entity replacement, and style transfer. Your task is to determine if the image and caption have been manipulated.*"

**Out-of-Context Classification (Task $T_2$):** "*Out-of-Context Classification is a task in which the goal is to identify whether a given image and accompanying text are contextually mismatched or falsely connected. Your task is to identify whether a given image and its accompanying text are contextually mismatched or falsely connected.*"

**Veracity Classification (Task $T_3$):** "*The Veracity task in a multimodal context involves assessing the truthfulness or accuracy of textual claims by using visual evidence. Your task is to determine the truthfulness of textual claims based on the accompanying visual evidence.*"

Besides, we carefully design three questions for the three MFC sub-tasks and incorporate the image $I$ and claim $C$ into them, to enable the model to answer questions for verdict prediction as follows:

**Manipulation Classification (Question $Q_1$):** "Given a claim $\{C\}$ and its image $\{I\}$, is this multimodal content manipulated?"

**Out-of-Context Classification (Question $Q_2$):** "Does this claim $\{C\}$ match its image $\{I\}$?"

**Veracity Classification (Question $Q_3$):** "Based on the image $\{I\}$, is this claim $\{C\}$ true?"

At the end of each prompt template, we instruct the required output format $F$: "Answer yes or no.". As demonstrated in Figure 2, to explore the effect of different prompt strategies like Chain-of-Thought (CoT) (Wei et al., 2022) or In-Context Learning (ICL) prompting, we utilized the four following prompt methods for the *MFC-Bench*: *Zero-shot*, *Zero-shot with CoT* (Kojima et al., 2022), *Few-shot*, and *Few-shot with CoT* (Wei et al., 2022). Specifically, we design the prompt as follows:

**Zero-shot Prompt.** We initially employed the zero-shot setting to activate the fact-checking capabilities of LVLMs. Given a task instruction $T_i$, a question unit $Q_i$, and the return format $F$, the

LVLMs $f(\cdot)$ are expected to determine whether the output $Y = f(T_i, Q_i, F)$ is "Yes" or "No", as depicted in Figure 2(a). To extend the Zero-shot with CoT setting in LLMs described in Kojima et al. (2022), we simply incorporated the CoT prompt $C_p$ "Let's think step by step" into the original prompt, to encourage the LVLMs to implicitly conduct complex reasoning by retrieving internal evidence, for determining the label $L$. Consequently, LVLMs will process $f(T_i, Q_i, C_p, F)$ and finally return the answer to multimodal fact verification.

**Few-shot Prompt.** Previous literature has indicated that pre-trained LLMs can significantly benefit from the inclusion of a few ICL demonstrations (Brown et al., 2020). To assess whether the LVLMs could gain similar advantages from the in-context demonstrations in multimodal fact-checking, we employed the few-shot setting. For the Few-shot examples, we define each example $E = \{Q_i, L\}$ consisting of a question $Q_i$ and its corresponding factuality label $L$ for fact verification. The inputs of LVLMs are given as $\{T_i, E^N, Q_i, F\}$, where $E^N$ represents multiple examples and $N$ denotes the number of examples, as demonstrated in Figure 2(b). In terms of the Few-shot with CoT prompt, we manually curated a rationale $R$ for each example to guide the LVLMs, where the example is represented as $E_c = \{Q_i, R, L\}$ and the input is $\{T_i, E_c^N, Q_i, F\}$.

**Justification Production** Furthermore, to gain deeper insights into the model interpretability of LVLMs, we expand our research on the evaluation of the justification production of LVLMs. The output format $F$: "Answer yes or no." was removed to allow the model to produce more intermediate reasoning steps. The model's interpretability was evaluated by GPT-4 and humans across four dimensions: Misleadingness (M), Informativeness (I), Soundness (S), and Readability (R). A 5-point Likert scale was used, where 1 indicates the lowest quality and 5 the highest for Informativeness, Soundness, and Readability, but the scale is reversed for Misleadingness. Detailed explanations of Misleadingness (M), Informativeness (I), Soundness (S), and Readability (R), as well as the prompts we used, can be found in Appendix §E.6.

## 4 EXPERIMENTS AND RESULTS

### 4.1 EXPERIMENTAL SETUP

We conduct extensive experiments on the *MFC-Bench* to evaluate a total of 18 representative LVLMs: 1) GPT-4o; 2) GPT-4V; 3) Claude3.5-Sonnet; 4) Claude3-Haiku; 5) Gemini-1.5-Pro; 6) Emu2; 7) InternVL; 8) CogVLM; 9) LLaVA-NeXT; 10) InstructBLIP; 11) Pixtral; 12) MiniCPM-V-2.6; 13) LLava-OneVsion; 14) Molmo; 15) Qwen-VL; 16) Qwen2-VL; 17) Yi-VL; 18) xGen-MM. To ensure our results are reproducible, we set the temperature as 0 without any sampling mechanism. We also have incorporated human performance as the benchmark baseline for comparison. We use the accuracy and macro-averaged F1 score (dominant) as the evaluation metrics. More implementation details and baseline descriptions are provided in Appendix §B-§C.

### 4.2 MAIN RESULTS

In Table 3, we present the average outcomes of the listed 18 accessible and representative LVLMs in a zero-shot setting on the *MFC-Bench*. From the results, we derive the following observations:

1) For the overall performance of the LVLMs on the Manipulation Classification, the proprietary model Gemini-1.5-Pro achieves the best performance with the 61.6% F1 score. In open-source models, Molmo performs the best, with the 59.3% F1 score. Counterintuitively, the more powerful closed-source models, namely GPT-4V, Claude3.5-Sonnet and Claude3-Haiku, fail to produce promising results in this sub-task. 2) None of the models exceeded the 62% F1 score, exposing weaknesses in vision-language models for this multimodal fact-checking stage. In contrast, human performance reached over 75%, indicating significant room for improvement in LVLMs. This discrepancy highlights that computational power alone does not ensure superior performance in Manipulation Classification. 3) In OOC Classification, GPT-4o stands out as the preeminent model with the highest 84.8% F1 score. In terms of Veracity Classification, Qwen2-VL is distinguished by its considerable F1 score of 75.5%. 4) Overall, we can find most of the LVLMs could achieve better performance on OOC Classification but worse on Manipulation Classification, and performance on Veracity Classification lies in the intermediate range. This pattern underscores the rational distribution of task difficulty within our proposed benchmark, *MFC-Bench*, which comprehensively spans a spectrum from challenging to straightforward multimodal fact-checking tasks. 5) In comparison to

Table 3: Results of different LVLMs on the *MFC-Bench*, in the zero-shot setting. The accuracy and macro-averaged F1 score (%) are reported as the metrics. The best and second test results are in bold and underlined, respectively.

| Models | Size | Manipulation | | OOC | | Veracity | | Overall | |
|---|---|---|---|---|---|---|---|---|---|
| | | Accuracy | F1 | Accuracy | F1 | Accuracy | F1 | Accuracy | F1 |
| *Proprietary Models* | | | | | | | | | |
| 🔷 GPT-4o | - | **65.7** | 60.4 | **84.8** | **84.8** | 80.1 | **63.0** | **67.7** | **69.4** |
| 🔷 GPT-4V | - | 58.4 | 50.2 | 75.8 | 75.2 | 77.4 | 60.0 | 60.6 | 61.8 |
| 🅰 Claude3.5-Sonnet | - | 59.9 | 41.7 | 49.9 | 37.6 | 72.7 | 47.4 | 60.1 | 42.2 |
| 🅰 Claude3-Haiku | - | 51.4 | 37.8 | 59.8 | 59.5 | **80.3** | 57.4 | 53.7 | 51.6 |
| 🟢 Gemini-1.5-Pro | - | 64.2 | **61.6** | 80.2 | 80.1 | 79.6 | 56.6 | 66.1 | 66.1 |
| *Open-Source Models* | | | | | | | | | |
| 〰 Emu2 | 37B | 38.7 | 33.0 | 51.9 | 51.1 | 70.0 | 52.6 | 41.4 | 45.6 |
| 🦌 InternVL | 25.5B | 60.1 | 44.6 | 73.4 | 73.0 | 80.0 | 57.4 | 62.1 | 58.3 |
| 🔵 CogVLM | 17B | 56.3 | 52.3 | 61.4 | 56.2 | 76.4 | 63.4 | 57.8 | 57.3 |
| 🖼 LLaVA-NeXT | 13B | **62.5** | 56.5 | 61.8 | 57.2 | 78.4 | 51.3 | 63.4 | 55.0 |
| ☁ InstructBLIP | 13B | 41.7 | 30.5 | 59.5 | 52.3 | 49.6 | 49.3 | 43.3 | 44.0 |
| 🅼 Pixtral | 12B | 58.5 | 43.9 | 64.8 | 63.5 | 80.9 | 65.0 | 60.2 | 57.5 |
| ≋ MiniCPM-V-2.6 | 8B | 58.9 | 39.7 | 71.2 | 71.0 | 80.4 | 65.1 | 60.9 | 58.6 |
| 🖼 LLaVA-OneVision | 7B | 61.5 | 55.5 | 75.7 | 75.4 | 80.9 | 60.3 | **63.5** | 63.7 |
| ✤ Molmo | 7B | 59.3 | **59.3** | 58.9 | 52.3 | 79.9 | 57.6 | 60.5 | 56.4 |
| 🔹 Qwen-VL | 7B | 45.7 | 45.4 | 69.7 | 69.4 | 82.7 | 69.3 | 49.4 | 61.4 |
| 🔹 Qwen2-VL | 7B | 59.9 | 46.6 | **80.1** | **80.1** | **85.7** | **75.5** | 62.7 | **67.4** |
| 🟡 Yi-VL | 6B | 56.4 | 43.8 | 70.4 | 70.4 | 78.4 | 60.0 | 58.6 | 58.1 |
| ☁ xGen-MM | 5B | 42.7 | 33.8 | 50.0 | 44.8 | 64.7 | 48.7 | 44.5 | 42.4 |
| *Human* | | | | | | | | | |
| 👤 Human | - | 75.7 | 75.6 | 74.0 | 73.5 | 96.0 | 91.7 | 76.8 | 80.3 |

Table 4: Justification Evaluated by GPT-4 and Human.

| Models | Misleadingness | | Informativeness | | Soundness | | Readability | |
|---|---|---|---|---|---|---|---|---|
| | GPT-4 | Human | GPT-4 | Human | GPT-4 | Human | GPT-4 | Human |
| 🖼 LLaVA-NeXT(7B) | 3.82 | 3.56 | 2.96 | 3.02 | 3.30 | 3.71 | 4.39 | 4.46 |
| 🖼 LLaVA-NeXT(13B) | 3.61 | 3.68 | 3.07 | 3.50 | 3.48 | 3.77 | 4.49 | 4.63 |
| ☁ InstructBLIP(7B) | 3.41 | 3.36 | 1.06 | 2.22 | 1.63 | 2.45 | 2.35 | 3.22 |
| ☁ InstructBLIP(13B) | 3.32 | 3.32 | 1.16 | 2.21 | 1.71 | 2.54 | 2.46 | 3.51 |
| 🔹 Qwen-VL | 3.76 | 3.61 | 1.77 | 2.63 | 2.63 | 3.11 | 3.68 | 3.64 |
| 🟡 Yi-VL | 3.04 | 3.30 | 2.04 | 2.34 | 3.31 | 3.56 | 4.20 | 4.50 |

humans, LVLMs show considerable potential for further development in addressing more complex fact-checking challenges like Manipulation Classification. Despite this, their performance is solid in simpler tasks like OOC Classification.

## 4.3 MODEL INTERPRETABILITY

We conducted a post-hoc interpretability analysis about Justification Production across six selected models: LLaVA-NeXT (7B&13B), InstructBLIP (7B&13B), Qwen-VL, and Yi-VL. This investigation explored the differences in justification production within the same model family yet varying parameter sizes, as well as the differences between distinct models. In Table 4, evaluations by GPT-4 and human evaluators show that the LLaVA-NeXT models perform exceptionally well, achieving high scores in Informativeness, Soundness, and Readability. In contrast, the InstructBLIP models struggle with interpretability. We speculate the reason is that the models are often limited to binary 'yes' or 'no' biased responses, and additional prompts fail to improve their explanatory capabilities. Additionally, an increase in the size of the LVLMs, from 7 billion to 13 billion parameters, correlates with enhanced interpretability, as observed in the improved metrics for both LLaVA-NeXT and

Table 5: Fleiss' Kappa ($\kappa$) scores for human evaluation of different models.

| Models | $\kappa$(M) | $\kappa$(I) | $\kappa$(S) | $\kappa$(R) |
|---|---|---|---|---|
| *Human Evaluation* | | | | |
| LLaVA-NeXT(7B) | 0.72 | 0.68 | 0.74 | 0.75 |
| LLaVA-NeXT(13B) | 0.70 | 0.69 | 0.76 | 0.77 |
| InstructBLIP(7B) | 0.65 | 0.60 | 0.67 | 0.70 |
| InstructBLIP(13B) | 0.63 | 0.58 | 0.65 | 0.72 |
| Qwen-VL | 0.71 | 0.66 | 0.72 | 0.74 |
| Yi-VL | 0.68 | 0.64 | 0.70 | 0.73 |

Table 6: Results of selected emerging LVLMs on the *MFC-Bench* with the zero-shot CoT setting.

| Models | Manipulation | | OOC | | Veracity | |
|---|---|---|---|---|---|---|
| | Acc. | F1 | Acc. | F1 | Acc. | F1 |
| *Proprietary Models* | | | | | | |
| GPT-4o | 65.8 | 59.6 | 67.6 | 65.0 | 77.6 | 51.9 |
| *Open-Source Models* | | | | | | |
| LLaVA-NeXT | 58.1 | 55.1 | 52.4 | 39.1 | 77.2 | 46.2 |
| InstructBLIP | 41.9 | 31.0 | 57.0 | 47.6 | 37.2 | 36.9 |
| LLaVA-OneVision | 61.2 | 58.3 | 73.3 | 72.7 | 81.3 | 61.6 |
| Qwen-VL | 45.7 | 45.2 | 71.9 | 71.8 | 81.8 | 65.3 |
| Qwen2-VL | 59.3 | 47.0 | 79.8 | 79.8 | 86.6 | 77.1 |
| Yi-VL | 59.9 | 42.5 | 69.4 | 69.3 | 78.0 | 56.1 |

InstructBLIP families. The Fleiss' Kappa ($\kappa$) scores shown in Table 5, reflects strong consistency among the annotators. More details of human evaluation and bias are in Appendix §E.5-§E.8.

## 4.4 EFFECT OF CoT

The comparison between Table 3 and Table 6 shows that the impact of CoT in the zero-shot setting varies across different selected representative LMMs on *MFC-Bench*. For Manipulation Classification, the impact of CoT on model performance differs, as seen in GPT-4o, where the F1 score decreases from 60.4% to 59.6%, and in LLaVA-OneVision, where it rises from 55.5% to 58.3%. In the case of OOC Classification, CoT proves beneficial for some LVLMs, such as Qwen-VL, while it negatively affects others, like Qwen2-VL. For Veracity Classification, CoT generally does not significantly impact performance and may even reduce it for certain models. In few-shot settings, as shown in Figure 3, CoT does not enhance the performance of LLaVA-OneVision and Qwen2-VL. For LLaVA-OneVision, CoT has a minimal to slightly positive impact on performance in Manipulation Classification and a somewhat negative impact in Veracity Classification. Conversely, the effect of CoT on the GPT-4o is continuously negative. The possible reasons for these observations include the underdeveloped ability of the LVLM to handle multiple image inputs and the excessive length of the rationale, which diminishes the model's ability to understand the task effectively.

## 4.5 EFFECT OF ICL

To thoroughly investigate the impact of In-Context Learning (ICL) on model performance, we selected GPT-4o, Qwen2-VL and LLaVA-OneVision that support multiple image inputs to conduct few-shot experiments. We calculated the macro-averaged F1 scores as the evaluation metric. 1) The results, as illustrated in Figure 3, indicate that the implementation of few-shot learning does not enhance the fact-checking capabilities of these models. 2) For the performance of Qwen2-VL in Figure 3, the few-shot prompt (i.e., ICL) did not result in a performance improvement. Instead, we found that it induced model inertia, leading it to predominantly respond with "no" in most instances. We provide more qualitative analysis in Appendix §E.

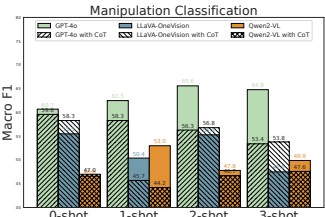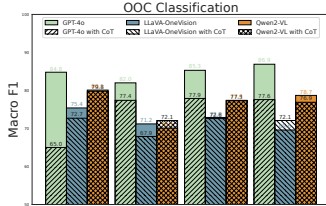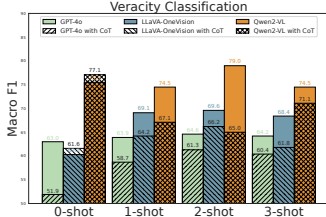

Figure 3: Comparison between few-shot conditions w/ and w/o CoT for GPT-4o, LLaVA-OneVision and Qwen2-VL.

## 5 ETHICS STATEMENT

The aim of this research is to focus on the multimodal fact-checking issue related to LVLMs, to curb the dissemination of multimodal disinformation, and to protect individuals from exposure to fake news. However, we acknowledge the risk that malicious actors might attempt to reverse-engineer misinformation that could evade detection by AI systems trained on LVLMs. We vehemently discourage and denounce such practices, and emphasize that human moderation is essential to prevent such occurrences. Our utilization of data adheres to the terms of the datasets (Shao et al., 2023; Luo et al., 2021; Yao et al., 2023). All the data in this work only includes text and image modalities and does not contain any user information on social media.

To protect our human evaluators, we establish three guidelines: 1) ensuring their acknowledgment of viewing potentially uncomfortable content, 2) limiting weekly evaluations and encouraging a lighter daily workload, and 3) advising them to stop if they feel overwhelmed. Finally, we regularly check in with evaluators to ensure their well-being.

## 6 CONCLUSION AND FUTURE WORK

In this study, we aim to investigate the trustworthy insight of LVLMs by examining the multimodal fact-checking ability of LVLMs across a spectrum of data categories. For this purpose, we have developed the *MFC-Bench*, a comprehensive testbed consisting of 35K multimodal samples, spanning three tasks of varied complexity. Our evaluation of various LVLMs using different prompting methods, including those with CoT or ICL prompts, on the *MFC-Bench* reveals that these models still exhibit limitations in accurately addressing multimodal fact-checking tasks. In our future work, we plan to systematically study justification production for multimodal fact-checking with LVLMs.

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

## A    DISTRIBUTION

The dataset is publicly available on the Hugging Face anonymous page: Manipulation Classification, OOC Classification and Veracity Classification.

The dataset is accompanied by Croissant metadata and licensing information all available on Hugging Face Hub.

## B    DESCRIPTIONS OF LVLM BASELINES

We conduct extensive experiments on the *MFC-Bench* to evaluate the following representative LVLMs:

- GPT-4o, the latest flagship model developed by OpenAI, designed for real-time reasoning across audio, visual, and textual inputs. It excels in understanding both vision and audio, offering significant improvements over previous models in these areas. We specifically utilize the "gpt-4o-2024-05-13" version.

- GPT-4V (OpenAI, 2023), developed by OpenAI, is a version of the GPT-4 architecture that includes capabilities for processing and generating images in addition to text. We specifically utilize the "gpt-4-vision-preview" version.

- Claude3.5-Sonnet developed by Anthropic with significant improvements most evident in visual reasoning tasks like interpreting charts and graphs, and it can accurately transcribe text from imperfect images We specifically utilize the "claude-3-5-sonnet-20240620" version.

- Claude3-Haiku[3], developed by Anthropic, possesses sophisticated vision capabilities comparable to other leading models. It can process a wide range of visual formats, including photos, charts, graphs, and technical diagrams. We specifically utilize the "claude-3-haiku-20240307" version.

- Gemini-1.5-Pro developed by google, can perform highly-sophisticated understanding and reasoning tasks for different modalities, including vision. We specifically utilize the "gemini-1.5-pro" version

- Emu2 (Sun et al., 2023) is a generative multimodal model with 37 billion parameters, designed to enhance task-agnostic in-context learning capabilities through effective scaling. We specifically utilize the "Emu2" version.

- InternVL (Chen et al., 2023c) is a large-scale vision-language foundation model, scaling up the vision foundation model to 6 billion parameters and progressively aligning it with the LLM, using web-scale image-text data from various sources. We specifically utilize the "InternVL-Chat-V1-5" version.

- CogVLM (Wang et al., 2023a) is a powerful open-source visual language foundation model that achieves state-of-the-art performance on multiple cross-modal benchmarks by using a trainable visual expert module for deep fusion of vision and language features. We specifically utilize the "cogvlm-chat" version.

---

[3]https://claude.ai/

- LLaVA-NeXT (Liu et al., 2024a) is the new version of LLaVA (Liu et al., 2023a), with improved reasoning, OCR, and world knowledge capabilities. We specifically utilize the "llava-v1.6-vicuna-7b, llava-v1.6-vicuna-13b, llava-v1.6-34b" version.

- InstructBLIP (Dai et al., 2023) introduces a novel vision-language instruction-tuning framework utilizing BLIP-2 models to enhance zero-shot generalization performance across diverse vision-language tasks. We specifically utilize the "instructblip-vicuna-7b, instructblip-vicuna-13b" version.

- Pixtral[4] developed by Mistral Ai, is trained to understand both natural images and documents, demonstrates strong abilities in tasks such as chart and figure understanding, document question answering, multimodal reasoning, and instruction following. We specifically utilize the "Pixtral-12B-2409" version.

- MiniCPM-V-2.6 (Yao et al., 2024) is the latest and most capable model in the MiniCPM-V series developed by OpenBMB, achieves an average score of 65.2 on the latest version of OpenCompass, a comprehensive evaluation over 8 popular benchmarks. We specifically utilize the "openbmb/MiniCPM-V-2_6" version.

- LLaVA-OneVision (Li et al., 2024a) is the first single model that can simultaneously push the performance boundaries of open LMMs in three important computer vision scenarios: single-image, multi-image, and video scenarios. We specifically utilize the "lmms-lab/llava-onevision-qwen2-7b-ov" version.

- Molmo (Deitke et al., 2024) developed by Allen Ai, is powerful model closes the gap between open and proprietary systems across a wide range of academic benchmarks as well as human evaluation. We specifically utilize the "allenai/Molmo-7B-D-0924" version.

- Qwen-VL (Bai et al., 2023) is Alibaba Cloud's multimodal large vision-language model that excels in multilingual text recognition, fine-grained understanding, and multi-image interleaved conversations, significantly outperforming other large vision-language models in various benchmarks. We specifically utilize the "Qwen/Qwen-VL-Chat" version.

- Qwen2-VL (Wang et al., 2024b) is the latest addition to the vision-language models in the Qwen series, building upon the capabilities of Qwen-VL. We specifically utilize the "Qwen/Qwen2-VL-7B-Instruct" version.

- mPLUG-Owl (Ye et al., 2023a), developed by DAMO Academy, is a training approach that enhances LLMs with multimodal capabilities by integrating a foundational LLM with a visual knowledge module and a visual abstractor module, using a two-stage method to align image and text. We specifically utilize the "mplug-owl-llama-7b" version.

- MiniGPT-v2 (Chen et al., 2023b) is a unified vision-language model designed for diverse tasks such as image description and visual question answering, utilizing unique task identifiers for improved performance and efficiency. We specifically built the model based on the "llama-2-7b-chat" LLaMA version with the checkpoint of the online developing demo.

- Yi-VL (Young et al., 2024) is an open-source multimodal vision-language model from the Yi LLM series, excelling in content comprehension and multi-round image conversations, and leading in recent English and Chinese benchmarks. We specifically utilize the "Yi-VL-6B" version.

- xGen-MM (Xue et al., 2024) is a series of the latest foundational Large Multimodal Models (LMMs) developed by Salesforce AI Research. This series advances upon the successful designs of the BLIP series, incorporating fundamental enhancements that ensure a more robust and superior foundation. We specifically utilize the "Salesforce/xgen-mm-phi3-mini-instruct-r-v1" version.

- MiniCPM-V-2[5] is a robust multimodal large language model designed for efficient end-side deployment. It is built on the foundation of SigLip-400M and MiniCPM-2.4B, connected by a perceiver resampler. We specifically utilize the "MiniCPM-V 2.0" version.

---

[4]https://mistral.ai/news/pixtral-12b/
[5]https://huggingface.co/openbmb/MiniCPM-V-2

## C    Implementation Details

The data processing for our datasets is centered around Figure 4, which handles both image and text data to construct the benchmark.

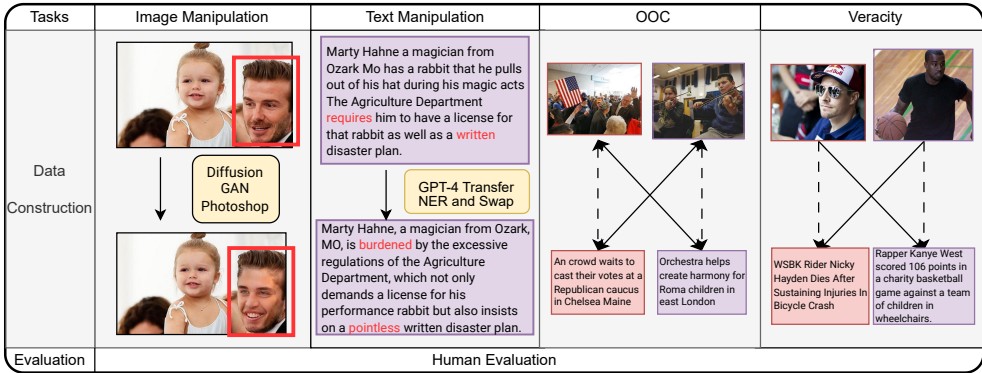

Figure 4: The pipeline of dataset construction.

### C.1    Data Construction

#### C.1.1    Manipulation Classification

To explore the potential capacity of LVLMs on Manipulation Classification in a multimodal context, we designed seven types of manipulation, selecting data from the DGM4 dataset (Shao et al., 2023) and constructing additional datasets ourselves. The initial data was sourced from the VisualNews (Liu et al., 2021) datasets. The DGM4 dataset complies with the Apache-2.0 license. The VisualNews dataset is available upon request.

- **Method 1: Face Swap (FS).** Face Swap involves the process of cutting a face from one image and replacing it with a different face in another image. It can be used to create realistic but fake images of public figures, such as politicians, celebrities, or journalists, appearing to do things they never did. It is important for LVLMs not only to verify the authenticity of news text content but also to accurately identify whether the individuals in the accompanying photos correspond to the reported events. We have sampled and chosen a Face Swap subset of the DGM4 dataset (Shao et al., 2023) as part of our benchmark to *detect Whether LVLMs can recognize public figures and retrieve information related to individuals from its internal parametric knowledge* through multimodal data.

  **Data processing**: A Face Swap subset of the DGM4 dataset (Shao et al., 2023) was sampled and selected.

- **Method 2: Face Attribute Edit (AE).** Unlike Face Swap, Face Attribute Edit achieves deception by altering the facial expressions of humans like newsmakers. This can be potentially harmful to the public, as it can particularly portray a public figure laughing inappropriately in a serious context, which is highly misleading and infuriating. To identify such discrepancies, LVLMs must precisely recognize the type of event and the expected demeanor of the individuals involved. Our benchmark randomly selected visual and textual samples related to face attribute editing from the previously established DGM4 dataset (Shao et al., 2023). This inclusion allows us to *evaluate the multimodal fact-checking capabilities of LVLMs in recognizing the scene, identifying personal information and detecting the correctness of face's status* in visual content in the multimodal context.

  **Data processing**: Visual and textual samples related to face attribute editing were randomly selected from the previously established DGM4 dataset (Shao et al., 2023).

- **Method 3: Background Change (BC).** The same individuals, involving the same events, can take place in different locations. Before the emergence of diffusion models, manip-

ulating a suitable scene was extremely challenging. However, with the advent of diffusion models (Rombach et al., 2022), we can now effortlessly alter the background of images, thereby creating scenes that did not originally exist in fact. Specifically, we are interested in whether LVLMs can exactly determine if the time and location of an event align with the actual scene. We utilized Grounding DINO (Liu et al., 2023b) and `stable-diffusion-inpainting`[6] models to generate a background for a manipulated and unrealistic outdoor scene. Our objective was to *examine the capability of LVLMs in faithfully identifying these artificially constructed counterfactual scenarios*.

**Data processing**: Backgrounds for outdoor scenes were generated using Grounding DINO (Liu et al., 2023b) and stable-diffusion-inpainting techniques. First, we used Grounding DINO to detect the people in the photos and create inverse masks. Then, we provided these masks along with the original images for stable-diffusion-inpainting. The pipeline was implemented using ComfyUI.

- **Method 4: CLIP-based Stable Diffusion Generate (CG).** Stable diffusion (SD) traditionally employs the text-to-image generation. However, by incorporating CLIP (Radford et al., 2021), we can transform the process into an image-to-image generation (Ramesh et al., 2022), enabling the manipulated image to retain the linguistic information from the original image. It is crucial for LVLMs to accurately discern between authentic and fabricated images by incorporating their internal knowledge, Using StabilityAI's `Stable-Diffusion-2-1-Unclip`[7], we generated stable diffusion versions of the original images for replacement. This design allows us to *test the fact-checking capacity of LVLMs for awareness of whether the multimodal contents have been manipulated with the original image information*.

  **Data processing**: Stable diffusion versions of the original images were generated using StabilityAI's Stable-Diffusion-2-1-Unclip. By utilizing Stable-Diffusion-2-1-Unclip, we input the original claim and image into the model to generate the manipulated images.

- **Method 5: Photoshop (PS).** Photoshop has long been a leading tool for manual image editing, enabling users to alter human figures and merge different images to create potentially misleading visuals. This capability can have serious consequences, as it may lead to the spread of misinformation, manipulate public perception, and distort reality. LVLMs must leverage their inherent knowledge, which encompasses a vast understanding of context, patterns, and nuances in visual data, to effectively identify and analyze such issues of manipulation and misinformation. This facilitates our assessment of *whether LVLMs can discern the traces of human manipulation, thereby fulfilling the requirements of the fact-checking task.*

  **Data processing**: To evaluate the effectiveness of LVLMs in detecting human manipulation, we utilize the photoshop subset of Fakeddit (Nakamura et al., 2020).

- **Method 6: Textual Entity Replace (ER).** Textual Entity Replace is a traditional method of text manipulation. Using Named Entity Recognition (NER) (Lample et al., 2016) from `bert-base-NER`[8], we identified named entities corresponding to persons within a given claim where newsmakers are mentioned. Subsequently, we randomly selected the location or time from an NER candidate set consisting of thousands of entities, to replace the target location or time entities in the claim. This creates counterfactual scenarios where the photos and claims contain the same individuals, but the events depicted are different. *This scenario challenges the ability of LVLMs to keenly associate individuals with events, relying on their internal factual knowledge.*

  **Data processing**: Named entities corresponding to persons within a given claim were identified using Named Entity Recognition (NER) (Lample et al., 2016) from bert-base-NER, and the surrounding contextual texts of the person would be replaced with other contexts of contradicted and different locations and time. To ensure that the claims contain people, we first screened the data and selected only the claims that included individuals.

- **Method 7: Text Style Transfer (ST).** Similar to Face Attribute Edit, Text Style Transfer can alter the perception of the same person and event, giving a different factual impres-

---

[6]https://huggingface.co/runwayml/stable-diffusion-inpainting
[7]https://huggingface.co/stabilityai/stable-diffusion-2-1-unclip
[8]https://huggingface.co/dslim/bert-base-NER

sion (Wu et al., 2024). For instance, an originally sad event can be described in a way that makes it seem humorous. This poses a substantial challenge for fact-checking efforts as it requires LVLMs not only to detect the factual content but also to understand the tone and style nuances that might misrepresent the underlying truth of the situation. Hence, we first utilized GPT-4 (OpenAI, 2023) to determine whether the sentiment of the text is positive or negative. Then, leveraging the advanced text style transfer capabilities of GPT-4, we rewrote the text to express the opposite sentiment. *The process examines LVLMs' ability to rigorously comprehend the events and associated sentiments depicted in images and claims, and to correctly correlate them.*

**Data processing**: The sentiment of the text was first determined using GPT-4 (OpenAI, 2023), and then the text was rewritten to express the opposite sentiment using GPT-4's advanced text style transfer capabilities.

### C.1.2    OUT-OF-CONTEXT CLASSIFICATION

Out-of-Context (OOC) Classification (Luo et al., 2021) aims to evaluate the coherence and correspondence of context across various modalities. Unlike the aforementioned manipulation techniques that require modifying images and texts, OOC Classification combines real but misused images and texts. If the image and claim are contextually aligned, we define the relationship as true. Conversely, if the image and claim are not contextually aligned, we define the relationship as false. We collected multimodal samples from the NewsCLIPpings dataset (Luo et al., 2021), using embedding methods such as CLIP and SBERT-WK (Wang & Kuo, 2020) to extract the most similar misused images, for *the evaluation of LVLMs' ability in discerning subtle semantic inconsistencies between images and texts* in OOC Classification.

**Data processing**: The Out-of-Context Classification data is sourced from the NewsCLIPpings (Luo et al., 2021) dataset. The NewsCLIPpings dataset is available upon request.

### C.1.3    VERACITY CLASSIFICATION

Veracity Classification (Yao et al., 2023) involves classifying the veracity of textual claims given retrieved visual evidence. Based on the image evidence, the LVLMs need to predict the truthfulness (Supported, Refuted) of the claim. We curated a subset of the Mocheg dataset (Yao et al., 2023) for this task. If the image supports the truthfulness of the claim, we label the relationship between the image and the claim as "Supported" indicating a true label. Otherwise, it is labeled as "Refuted" indicating a false label. This is a cross-modal semantic transformation task designed to *test whether LVLMs can accurately interpret and analyze visual information to support or refute textual claims*.

**Data processing**: the Veracity Classification data is obtained and sampled randomly from the Mocheg dataset (Yao et al., 2023). Mocheg dataset complies with the Apache-2.0 license.

### C.2    QUALITY ASSURANCE

This research involved a human subjects study to evaluate the quality of multimodal data manipulated by our adopted techniques. To assure the quality of the self-constructed data, we employed three human evaluators, who are senior undergraduate or graduate students majoring in computer science. Each student is presented with the manipulated data and the original data to judge whether the data has been successfully manipulated with the manipulation techniques for the reliability and credibility of the multimodal data. Each evaluator completes the quality assurance process independently.

The following considerations were adhered to ensure the protection and ethical treatment of participants: 1) Voluntary Participation: All participants were informed about the nature of the research and their role in it. Participation was entirely voluntary, with participants having the right to withdraw at any time without any consequences. 2) Informed Consent: Written informed consent was obtained from all participants. This consent form detailed the purpose of the research, the procedures involved, potential risks, and measures taken to safeguard participant data. 3) Data Anonymity and Confidentiality: All data collected during the study were anonymized. Personal identifiers were removed to maintain confidentiality and data were stored securely to prevent unauthorized access.

4) Minimal Risk: The study involved minimal risk to participants. The tasks performed were similar to everyday activities, and no sensitive personal information was requested or recorded.

## C.3  BENCHMARK COMPARISON

As shown in Table 7, our benchmark includes more comprehensive data and covers a wider range of sub-tasks in multimodal fact-checking. Our dataset consists of three types of tasks and nine specific data categories.

Table 7: Comparison of datasets related to multimodal fact-checking.

| Datasets | Manipulation | | | | | | | OOC | Veracity |
| --- | --- | --- | --- | --- | --- | --- | --- | --- | --- |
| | FS | AE | BC | CG | PS | ER | ST | | |
| Fakeddit (Nakamura et al., 2020) | ✗ | ✗ | ✗ | ✗ | ✔ | ✗ | ✗ | ✔ | ✗ |
| DGM4 (Shao et al., 2023) | ✔ | ✔ | ✗ | ✗ | ✗ | ✗ | ✗ | ✗ | ✗ |
| MEIR (Sabir et al., 2018) | ✗ | ✗ | ✗ | ✗ | ✔ | ✔ | ✗ | ✗ | ✗ |
| EMU (Da et al., 2021) | ✗ | ✗ | ✗ | ✗ | ✔ | ✗ | ✗ | ✗ | ✗ |
| Mocheg (Yao et al., 2023) | ✗ | ✗ | ✗ | ✗ | ✗ | ✗ | ✗ | ✗ | ✔ |
| NewsCLIPpings (Luo et al., 2021) | ✗ | ✗ | ✗ | ✗ | ✗ | ✗ | ✗ | ✔ | ✗ |
| MAIM (Jaiswal et al., 2017) | ✗ | ✗ | ✗ | ✗ | ✗ | ✗ | ✗ | ✔ | ✗ |
| COSMOS (Aneja et al., 2023) | ✗ | ✗ | ✗ | ✗ | ✗ | ✗ | ✗ | ✔ | ✗ |
| MMFakeBench (Liu et al., 2024b) | ✗ | ✗ | ✗ | ✔ | ✔ | ✔ | ✗ | ✔ | ✔ |
| *MFC-Bench* | ✔ | ✔ | ✔ | ✔ | ✔ | ✔ | ✔ | ✔ | ✔ |

## C.4  GPUs USAGE

We utilized the high-performance computing platform and employed Slurm to request 2-4 A800 GPUs for benchmarking multimodal fact-checking with LVLMs.

# D  RELATED WORK

## D.1  LLMs AND LVLMs

Recent advancements have seen LLMs excel across various domains, with major tech companies developing high-performing proprietary models such as OpenAI's GPT-3 (Brown et al., 2020) and GPT-4 (OpenAI, 2023), Google's PaLM (Chowdhery et al., 2022) and Gemini (Team et al., 2023), and Anthropic's Claude. These models, however, are often only accessible via specific APIs or not at all. In contrast, the AI community has embraced the emergence of open-source LLMs, making significant contributions like MistralAI's Mistral-series (Jiang et al., 2023), Google's UL2-20B (Tay et al., 2023) and Gemma (Mesnard et al., 2024), Tsinghua University's GLM-130B (Zeng et al., 2023), and Meta's OPT (Zhang et al., 2022) and the LLaMA series (Touvron et al., 2023a;b; Meta, 2024), enhanced by extensive alignment efforts (Wang et al., 2023c; Xu et al., 2023; Luo et al., 2023b;a; Mukherjee et al., 2023; Zhou et al., 2023; Li et al., 2023b).

LVLMs have significantly advanced the understanding of both textual and visual data within a unified framework (Chen et al., 2023a; Zhang et al., 2024). Innovative models such as Flamingo (Alayrac et al., 2022) and PaLM-E (Driess et al., 2023) have demonstrated the ability to integrate visual and textual information effectively, without the need for task-specific training. Concurrently, the development of diverse multimodal datasets (Yang et al., 2023) stemming from GPT-4 and GPT-4V (OpenAI, 2023) has spurred the fine-tuning of models like LLaVA (Liu et al., 2023a), MiniGPT-4 (Zhu et al., 2023), mPLUG-Owl (Ye et al., 2023b), InstructBLIP (Dai et al., 2023), and others (Bai et al., 2023; Wang et al., 2023b; Gong et al., 2023; Team et al., 2023; Bavishi et al., 2023), highlighting a trend towards more versatile and real-world applicable multimodal systems.

## D.2 Factual Knowledge in LMs

Previous studies have established that language models (LMs) can function as repositories of factual knowledge, serving effectively as knowledge bases (Petroni et al., 2019; 2020; Heinzerling & Inui, 2021). This reservoir of factual information acquired during pretraining proves beneficial for knowledge-intensive tasks, such as question-answering and fact-checking (Roberts et al., 2020; Yu et al., 2022; Pan et al., 2023). Petroni et al. (2019) used cloze tests involving triples and tailored prompts to evaluate the factual knowledge embedded in language models, while Jiang et al. (2020) focused on optimizing prompt design to enhance factual retrieval from these models.

Despite these advancements, the reliability of these methods has been questioned. Elazar et al. (2021) highlighted the inconsistency in rank-based probing methods when using paraphrased contexts. Similarly, Cao et al. (2021) argued that biased prompting and the leakage of correct answers can often lead to an overestimation of LM's knowledge retention. On the other hand, Varshney et al. (2022) employed question-answering formats to gauge models' uncertainty about specific facts, suggesting a different approach to measure factual accuracy. Our methodology aligns more closely with the approaches of Kadavath et al. (2022); Lin et al. (2022b); Hu et al. (2024), which involve querying models directly to self-evaluate their accuracy in delivering factual responses, offering a more direct assessment of their knowledge capabilities. But differently, this work focuses on the multimodal nature of fact checking to explore the complex reasoning capability of LVLMs.

## D.3 Multimodal Fact-Checking

Multimodal Fact-Checking refers to the systematic process of identifying counterfactuals or inconsistencies between facts across different modalities within multimodal data (Akhtar et al., 2023). Common manifestations of multimodal misinformation include claims about digitally manipulated context (Agarwal et al., 2019; Shao et al., 2023) and the amalgamation of context from disparate modalities and contexts (Luo et al., 2021; Aneja et al., 2021). The former is predominantly associated with deepfake technologies (Maras & Alexandrou, 2018; Dolhansky et al., 2019), while the latter is linked with cheapfake methodologies (Aneja et al., 2021). An essential Multimodal Fact-Checking pipeline consists of evidence retrieval and the adjudication process. Evidence retrieval furnishes the foundational basis for subsequent multimodal judgments. Within the adjudication phase, tasks are delineated into distinct categories, such as Manipulation Classification, Out-of-Context Classification, and Veracity Classification.

Manipulation Classification (Shao et al., 2023) is a task meticulously designed to ascertain whether multimodal data encompasses fabricated elements. Out-of-Context Classification (Luo et al., 2021) aims to evaluate the coherence and correspondence of context across various modalities. Veracity Classification (Yao et al., 2023) involves assessing whether the context from one modality aligns with or accurately reflects the context from another modality. Collectively, these tasks constitute the comprehensive process of multimodal fact-checking. In this work, we employed six different manipulation techniques to assess whether LVLMs can detect manipulations in multimodal news. Data from the NewsCLIPpings dataset is used to challenge LVLMs' ability to discern semantic differences between real images and real text, specifically for OOC classification. Similar to text, the cross-modal Veracity task is used to evaluate LVLMs' ability to perform factual inference across different modalities.

## D.4 Benchmarks for LVLMs

Traditional multimodal benchmarks have been centered around specific skills such as visual recognition (Goyal et al., 2017), image description (Agrawal et al., 2019), and visual commonsense reasoning (Zellers et al., 2019). However, the advent of advanced LVLMs has necessitated the development of new benchmarks to keep pace with their robust zero-shot capabilities, which often exceed those measured by conventional metrics. This has exposed shortcomings in their ability to match answers accurately, highlighting issues with robustness. To address these limitations, the research community has introduced several innovative benchmarks, such as MME (Fu et al., 2023), MMBench (Liu et al., 2023c), MM-Vet (Yu et al., 2023), SEED-Bench (Li et al., 2023a), GOAT-Bench (Lin et al., 2024b), LAMM (Yin et al., 2023) and MMCode (Li et al., 2024b). These benchmarks are designed to facilitate structured evaluations of complex multimodal tasks and reveal the flaws of traditional methods. Distinct from these, our proposed benchmark is tailored to systematically assess multi-

Table 8: Results of different LVLMs on the *MFC-Bench*, in the zero-shot setting. The accuracy and macro-averaged F1 score (%) are reported as the metrics.

| Models | Size | Manipulation | | OOC | | Veracity | |
|---|---|---|---|---|---|---|---|
| | | Accuracy | F1 | Accuracy | F1 | Accuracy | F1 |
| *Proprietary Models* | | | | | | | |
| 🆂 **GPT-4o** | - | 65.7 | 60.4 | **84.8** | **84.8** | 80.1 | 63.0 |
| 🆂 **GPT-4V** | - | 58.4 | 50.2 | 75.8 | 75.2 | 77.4 | 60.0 |
| Ⓐ **Claude3.5-Sonnet** | - | 59.9 | 41.7 | 49.9 | 37.6 | 72.7 | 47.4 |
| Ⓐ **Claude3-Haiku** | - | 51.4 | 37.8 | 59.8 | 59.5 | 80.3 | 57.4 |
| Ⓖ **Gemini-1.5-Pro** | - | 57.7 | 36.6 | 80.2 | 80.1 | 79.6 | 56.6 |
| *Open-Source Models* | | | | | | | |
| 🔯 **Emu2** | 37B | 38.7 | 33.0 | 51.9 | 51.1 | 70.0 | 52.6 |
| 🦙 **InternVL** | 25.5B | 60.1 | 44.6 | 73.4 | 73.0 | 80.0 | 57.4 |
| ◉ **CogVLM** | 17B | 56.3 | 52.3 | 61.4 | 56.2 | 76.4 | 63.4 |
| 🖼 **LLaVA-NeXT** | 13B | 62.5 | 56.5 | 61.8 | 57.2 | 78.4 | 51.3 |
| 🔵 **InstructBLIP** | 13B | 41.7 | 30.5 | 59.5 | 52.3 | 49.6 | 49.3 |
| 🅼 **Pixtral** | 12B | 58.5 | 43.9 | 64.8 | 63.5 | 80.9 | 65.0 |
| 🗇 **MiniCPM-V-2.6** | 8B | 58.9 | 39.7 | 71.2 | 71.0 | 80.4 | 65.1 |
| 🖼 **LLaVA-OneVision** | 7B | 61.5 | 55.5 | 75.7 | 75.4 | 80.9 | 60.3 |
| ✴ **Molmo** | 7B | 59.3 | 59.3 | 58.9 | 52.3 | 79.9 | 57.6 |
| 🐦 **Qwen-VL** | 7B | 45.7 | 45.4 | 69.7 | 69.4 | 82.7 | 69.3 |
| 🐦 **Qwen2-VL** | 7B | 59.9 | 46.6 | 80.1 | 80.1 | 85.7 | 75.5 |
| 🐵 **mPLUG-Owl** | 7B | 45.7 | 45.4 | 48.3 | 46.1 | 60.8 | 49.7 |
| 🅨 **Yi-VL** | 6B | 56.4 | 43.8 | 70.4 | 70.4 | 78.4 | 60.0 |
| ☁ **xGen-MM** | 5B | 42.7 | 33.8 | 50.0 | 44.8 | 64.7 | 48.7 |
| 🗇 **MiniCPM-V-2** | 2.8B | 64.0 | 56.6 | 67.2 | 66.3 | 81.8 | 65.5 |
| *Human* | | | | | | | |
| 👤 **Human** | - | **75.7** | **75.6** | 74.0 | 73.5 | **96.0** | **91.7** |

modal factual knowledge, especially concerning disinformation detection in the realm of deepfakes and cheapfakes. This testbed would allow for a more thorough exploration of LVLMs' trustworthy awareness concerning a wider range of task types associated with multimodal factuality.

# E   MORE RESULTS AND ANALYSIS

## E.1   ZERO-SHOT EVALUATION RESULTS

Table 8 shows the zero-shot evaluation results of a total of 20 LVLMs on the *MFC-Bench* in the zero-shot setting.

## E.2   POTENTIAL TEST SET LEAKAGE

For the open-source LVLMs, test set leakage is not a concern, as the literature explicitly delineates the datasets and instruction-tuning procedures employed in their training, none of which encompass the multimodal data utilized in our *MFC-Bench*. However, we cannot fully guarantee the exclusion of potential data leakage with the proprietary models, as its internal workings remain opaque. Nevertheless, as evidenced by the results in the experiments, where all LVLMs were evaluated directly on the *MFC-Bench*, the absence of significant test set leakage is implied. This is inferred from the fact that direct application of the LVLMs did not yield disproportionately high performance, which would be expected if the models were benefiting from test set leakage.

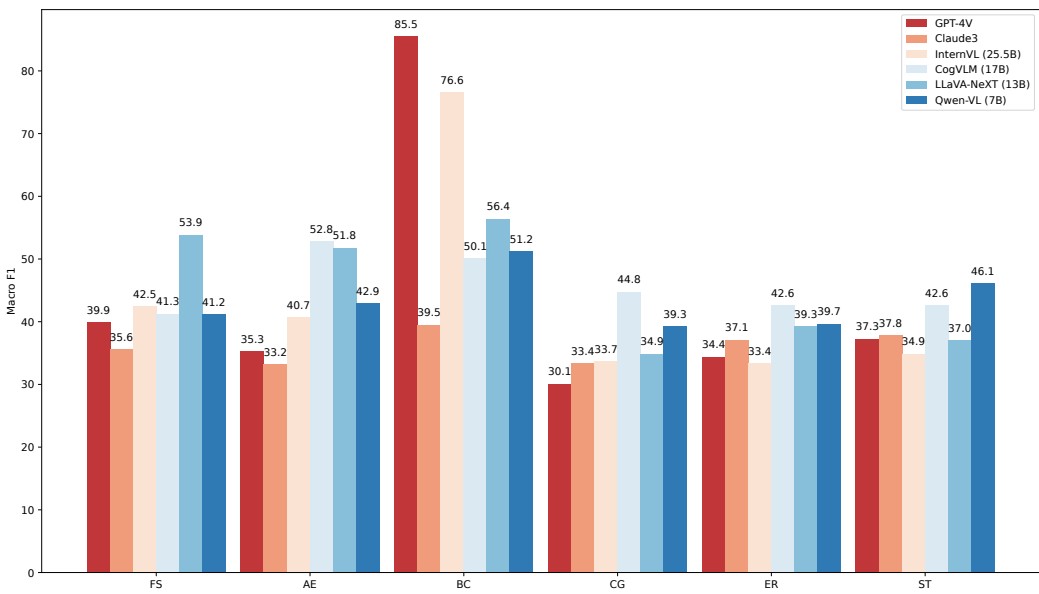

Figure 5: Effect of prompts specifically designed for different types of manipulation techniques.

## E.3 RESULTS ON DIFFERENT MANIPULATION TECHNIQUES

We further provide the detailed results of the representative LVLMs on the Manipulation Classification with respect to the seven manipulation methods, as depicted in Table 9.

Table 9: Detailed results of LVLMs on the Manipulation Classification in the zero-shot setting.

| Models | Size | FS | | AE | | BC | | CG | | PS | | ER | | ST | |
|---|---|---|---|---|---|---|---|---|---|---|---|---|---|---|---|
| | | Acc. | F1 | Acc. | F1 | Acc. | F1 | Acc. | F1 | Acc. | F1 | Acc. | F1 | Acc. | F1 |
| *Proprietary Models* | | | | | | | | | | | | | | | |
| GPT-4o | - | 61.4 | 45.7 | 60.8 | 42.9 | 78.6 | 73.2 | 63.6 | 60.8 | 80.4 | 80.4 | 58.1 | 53.7 | 56.8 | 49.5 |
| GPT-4V | - | 52.5 | 40.7 | 49.5 | 37.1 | 82.2 | 81.3 | 52.3 | 44.6 | 77.3 | 77.2 | 47.5 | 36.3 | 47.3 | 34.2 |
| Claude3.5-Sonnet | - | 62.7 | 41.0 | 64.4 | 39.7 | 69.0 | 47.6 | 53.7 | 36.9 | 59.3 | 49.2 | 58.7 | 38.8 | 51.0 | 35.5 |
| Claude3-Haiku | - | 50.2 | 35.8 | 50.2 | 36.1 | 50.0 | 35.5 | 50.2 | 35.7 | 51.4 | 42.3 | 57.4 | 42.3 | 50.7 | 37.2 |
| Gemini-1.5-Pro | - | 63.2 | 49.1 | 62.8 | 47.3 | 77.8 | 71.1 | 54.4 | 45.5 | 84.3 | 84.3 | 61.2 | 51.0 | 56.8 | 48.3 |
| *Open-Source Models* | | | | | | | | | | | | | | | |
| Emu2 | 37B | 35.5 | 30.7 | 35.3 | 30.0 | 32.7 | 25.9 | 33.6 | 28.8 | 57.3 | 52.6 | 57.7 | 42.6 | 49.8 | 38.1 |
| InternVL | 25.5B | 64.4 | 44.4 | 65.1 | 43.9 | 78.9 | 71.3 | 53.0 | 41.5 | 52.1 | 39.4 | 57.8 | 37.0 | 50.5 | 36.2 |
| CogVLM | 17B | 54.0 | 51.6 | 53.1 | 50.4 | 71.7 | 70.5 | 60.7 | 58.9 | 50.0 | 33.4 | 41.9 | 29.5 | 48.2 | 41.1 |
| LLaVA-NeXT | 13B | 60.7 | 51.2 | 60.5 | 50.7 | 81.8 | 79.9 | 61.9 | 59.6 | 63.5 | 59.9 | 54.2 | 41.5 | 55.5 | 51.2 |
| InstructBLIP | 13B | 33.6 | 25.7 | 33.6 | 25.8 | 33.6 | 25.7 | 50.5 | 35.8 | 49.1 | 33.4 | 42.2 | 30.9 | 50.7 | 36.7 |
| Pixtral | 12B | 64.4 | 44.9 | 64.5 | 44.9 | 66.9 | 50.5 | 50.5 | 38.7 | 57.3 | 52.7 | 57.2 | 42.5 | 52.0 | 41.1 |
| MiniCPM-V-2.6 | 8B | 66.2 | 41.6 | 66.3 | 42.0 | 68.1 | 45.8 | 50.4 | 35.5 | 54.0 | 43.3 | 57.6 | 37.5 | 49.9 | 34.2 |
| LLaVA-OneVision | 7B | 59.9 | 51.3 | 58.7 | 49.9 | 78.5 | 73.0 | 60.9 | 56.2 | 71.6 | 71.0 | 55.2 | 37.9 | 48.2 | 35.1 |
| Molmo | 7B | 51.4 | 50.2 | 52.3 | 51.0 | 64.6 | 64.3 | 70.4 | 69.8 | 61.4 | 56.0 | 47.1 | 45.9 | 51.2 | 51.1 |
| Qwen-VL | 7B | 45.4 | 45.2 | 46.3 | 46.1 | 46.9 | 46.8 | 46.9 | 46.2 | 41.6 | 41.6 | 47.2 | 46.4 | 40.2 | 40.0 |
| Qwen2-VL | 7B | 64.8 | 45.5 | 64.7 | 44.7 | 74.5 | 64.5 | 51.0 | 37.9 | 65.8 | 65.7 | 55.5 | 37.6 | 51.7 | 39.0 |
| mPLUG-Owl | 7B | 45.5 | 45.5 | 45.1 | 45.1 | 47.7 | 47.7 | 50.5 | 49.4 | 47.1 | 46.2 | 50.3 | 44.7 | 49.2 | 48.2 |
| Yi-VL | 6B | 65.3 | 44.2 | 64.7 | 43.7 | 68.9 | 50.5 | 51.2 | 40.2 | 64.7 | 63.5 | 56.4 | 37.4 | 49.6 | 36.8 |
| xGen-MM | 5B | 35.3 | 29.6 | 35.4 | 29.7 | 35.1 | 29.5 | 49.9 | 36.5 | 50.0 | 33.6 | 48.4 | 43.0 | 49.5 | 36.3 |
| MiniCPM-V-2 | 2.8B | 62.2 | 50.4 | 62.5 | 50.1 | 83.7 | 85.8 | 63.1 | 59.9 | 70.7 | 70.2 | 56.8 | 39.2 | 49.6 | 38.9 |
| *Human* | | | | | | | | | | | | | | | |
| Human | - | 63.0 | 62.9 | 71.0 | 70.9 | 92.0 | 92.0 | 91.0 | 91.0 | 75.9 | 75.4 | 59.0 | 58.8 | 78.0 | 77.9 |

## E.4 EFFECT OF PROMPTS ON MANIPULATION CLASSIFICATION

To verify the model's understanding of manipulation data, we designed prompts for six different manipulation methods and tested them on twelve models (see §F). As shown in Figure 5, the model's performance on each sub-task was consistent with that of a single prompt. This suggests that the

Table 10: Results of human evaluation on the *MFC-Bench* across different multimodal fact-checking tasks in a zero-shot setting.

| Tasks | Accuracy | F1 |
|---|---|---|
| Manipulation Classification | 75.67 | 75.58 |
| OOC Classification | 74.00 | 73.50 |
| Veracity Classification | 96.00 | 91.70 |

Table 11: Detailed results of human evaluation on the Manipulation Classification in the zero-shot setting.

| Tasks | Accuracy | F1 |
|---|---|---|
| FS | 63.0 | 62.9 |
| AE | 71.0 | 70.9 |
| BC | 92.0 | 92.0 |
| CG | 91.0 | 91.0 |
| PS | 75.9 | 75.4 |
| ER | 59.0 | 58.8 |
| ST | 78.0 | 77.9 |

model struggles with manipulation fact-checking. For the Background Change task, the scenarios we set might have been too simple, making it easy for the model to detect the manipulations.

## E.5 HUMAN EVALUATION

To assess the effectiveness of the *MFC-Bench* and better evaluate the performance of LVLMs, we conducted human evaluation experiments. For each sub-task, as illustrated in Figure 1, we randomly selected 100 samples, resulting in a total of 900 examples for human evaluation. 3 professional fact-checking annotators (between the ages of 26 and 29) were asked to judge the truthfulness of each sample (i.e., "Fact." or "Non-Fact.") in the zero-shot evaluation setting. Then the voting results were regarded as the answers. The results from their votes were then considered as the final evaluation.

As demonstrated in Table 10 and Table 11: 1) The accuracy of human predictions significantly surpasses LVLMs in Manipulation Classification. Humans achieved an accuracy of 75.67% and an F1 score of 75.58%. In Background Change and CLIP-based Stable Diffusion Generation methods, human accuracy exceeded 90%. Human fact-checking ability in Manipulation Classification surpasses that of LVLMs, suggesting that there is considerable room for improvement in LVLM performance. 2) Human performance in OOC classification is on par with the best-performing LVLMs, such as GPT-4V. Without manipulating the text and image, LVLMs can effectively identify the false connections between them. 3) For Veracity Classification, humans achieved an accuracy of over 95%. This high accuracy can be attributed to two factors: the strong fact-checking abilities of humans and the high degree of correlation within the dataset, which allowed humans to draw on their experience.

Human performance exceeds that of most LVLMs, especially in Manipulation Classification. This indicates that there is still significant potential for improvement in the fact-checking capabilities of LVLMs.

## E.6 MODEL INTERPRETABILITY

To gain deeper insights into the model interpretability of LVLMs, we expand our research on the evaluation of the justification production of LVLMs. The output format $F$: "Answer yes or no." was removed to allow the model to produce more intermediate reasoning steps.

For the evaluation of justification production, traditional automated evaluation metrics are inadequate to assess the output results of LVLMs (Chang et al., 2024). Fortunately, GPT-4 has been demonstrated to excel in assessing text quality from multiple angles, even in the absence of reference texts (Lin et al., 2024a; Wang et al., 2024a). Thus the model's justification was evaluated

Table 12: Model Interpretability Evaluated by GPT-4 and Human.

| Models | Size | Manipulation | | | | OOC | | | | Veracity | | | |
|---|---|---|---|---|---|---|---|---|---|---|---|---|---|
| | | M | I | S | R | M | I | S | R | M | I | S | R |
| *Evaluated by GPT-4* | | | | | | | | | | | | | |
| LLaVA-NeXT(7B) | 7B | 3.95 | 3.09 | 3.24 | 4.39 | 3.82 | 3.09 | 3.54 | 4.56 | 3.68 | 2.69 | 3.12 | 4.22 |
| LLaVA-NeXT(13B) | 13B | 3.83 | 3.16 | 3.36 | 4.46 | 3.57 | 3.17 | 3.70 | 4.61 | 3.44 | 2.89 | 3.39 | 4.41 |
| InstructBLIP(7B) | 7B | 3.86 | 1.06 | 1.47 | 2.24 | 3.04 | 1.11 | 1.87 | 2.60 | 3.32 | 1.00 | 1.54 | 2.21 |
| InstructBLIP(13B) | 13B | 3.67 | 1.42 | 1.92 | 2.71 | 2.88 | 1.06 | 1.69 | 2.44 | 3.42 | 1.00 | 1.53 | 2.23 |
| Qwen-VL | 7B | 4.02 | 1.83 | 2.61 | 3.73 | 3.82 | 1.64 | 2.45 | 3.47 | 3.43 | 1.85 | 2.83 | 3.85 |
| Yi-VL | 6B | 3.44 | 2.18 | 3.20 | 4.20 | 3.02 | 2.12 | 3.35 | 4.23 | 2.65 | 1.82 | 3.39 | 4.16 |
| *Evaluated by Human* | | | | | | | | | | | | | |
| LLaVA-NeXT(7B) | 7B | 3.43 | 3.15 | 3.83 | 4.47 | 3.82 | 2.09 | 3.54 | 4.56 | 3.42 | 3.82 | 3.76 | 4.34 |
| LLaVA-NeXT(13B) | 13B | 3.63 | 3.43 | 3.96 | 4.87 | 3.57 | 3.17 | 3.70 | 4.61 | 3.83 | 3.89 | 3.64 | 4.42 |
| InstructBLIP(7B) | 7B | 3.80 | 2.13 | 2.41 | 2.63 | 3.04 | 2.11 | 2.87 | 3.45 | 3.25 | 2.41 | 2.06 | 3.57 |
| InstructBLIP(13B) | 13B | 3.78 | 2.17 | 2.83 | 2.76 | 2.88 | 2.06 | 2.69 | 3.95 | 3.30 | 2.40 | 2.11 | 3.83 |
| Qwen-VL | 7B | 3.46 | 2.74 | 3.52 | 3.13 | 3.45 | 2.20 | 2.45 | 3.47 | 3.91 | 2.96 | 3.35 | 4.31 |
| Yi-VL | 6B | 3.54 | 2.53 | 3.81 | 4.56 | 3.23 | 2.20 | 3.35 | 4.23 | 3.14 | 2.28 | 3.52 | 4.72 |

by GPT-4 and Human subjects across four dimensions: Misleadingness (M), Informativeness (I), Soundness (S), and Readability (R). A 5-point Likert scale was used, where 1 indicates the lowest quality and 5 the highest for Informativeness, Soundness, and Readability, but the scale is reversed for Misleadingness.

- **Misleadingness (M)** assesses whether the model's explanation is consistent with the real veracity label of a claim, with a rating scale ranging from 1 (not misleading) to 5 (very misleading).

- **Informativeness (I)** measures how much the explanation provides new information, such as explaining the background and additional context, with a rating scale ranging from 1 (not informative) to 5 (very informative).

- **Soundness (S)** describes whether the explanation seems valid and logical, with a rating scale ranging from 1 (not sound) to 5 (very sound).

- **Readability (R)** evaluates whether the explanation follows proper grammar and structural rules, and whether the sentences in the explanation fit together and are easy to follow with a rating scale ranging from 1 (not fluent) to 5 (very fluent).

To use GPT-4 to evaluate the model interpretability of LVLMs, we carefully designed the following prompt. First, we give the GPT-4 system prompt "*You are now the judge of the model output.*"; Next, we provide GPT-4 with both the label $L$ and model output $Y$ using the format "Label:$\{L\}$, Model output $\{Y\}$". Finally, GPT-4 evaluates the output in four dimensions and returns in JSON format. Below is the complete prompt we use for GPT-4:

*Label:$\{L\}$*

*Model output: $\{Y\}$*

*Please rate in four dimensions*:

*1. Misleadingness -assesses whether the model's explanation is consistent with the real veracity label of a claim, with a rating scale ranging from 1 (not misleading) to 5 (very misleading)*

*2. Informativeness - assesses whether the explanation provides new information, such as explaining the background and additional context, with a rating scale ranging from 1 (not informative) to 5 (very informative)*

*3. Soundness - describes whether the explanation seems valid and logical, with a rating scale ranging from 1 (not sound) to 5 (very sound)*

*4. Readability - evaluates whether the explanation follows proper grammar and structural rules, and whether the sentences in the explanation fit together and are easy to follow with a rating scale ranging from 1 (poor) to 5 (excellent).*

*Scores 1-5, returned in json format.*

We conducted model interpretability analysis across six models: LLaVA-NeXT (7B), LLaVA-NeXT (13B), InstructBLIP (7B), InstructBLIP (13B), Qwen-VL, and Yi-VL. This investigation explored the differences within the same model family with varying parameter sizes, as well as the differences between distinct models. For the human subject study, the 3 professional annotators were asked to judge the model interpretability. The Fleiss' Kappa ($\kappa$) scores are shown in Table 5. Moreover, the intra-class agreement score is 0.685. The results are shown in Table 12.

### E.7 EFFECT OF MODEL SIZE

To explore the impact of model size on factual capabilities, we analyzed two families of LVLMs: InstructBLIP and LLaVA-NeXT, which both utilize the same language backbone, i.e., Vicuna (Chiang et al., 2023), and employ similar CLIP models, with InstructBLIP using EVA CLIP-g and LLava-NeXT using CLIP ViT-L/14. Specifically, we examined InstructBLIP (7B), InstructBLIP (13B), LLava-NeXT (7B), LLava-NeXT (13B), and LLava-NeXT (34B). As shown in Figure 6, the following observations were made: 1) In Manipulation Classification, there is a minimal correlation between the model size of the specific LVLMs family and the performance. 2) Regarding OOC Classification and Veracity Classification, the model performance generally improves with the increased model size.

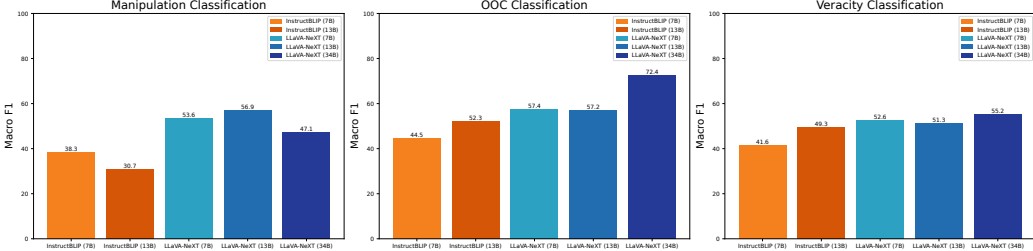

Figure 6: Model size effects of LVLMs.

### E.8 YES/NO BIAS

During benchmarking, we identified a Yes/No Bias issue with the tested LVLMs, where it tends to consistently respond with either "yes" or "no". We have chosen two key metrics to evaluate the Yes/No bias of the model for the Manipulation Classification task: 1) False Positive Rate (FPR) (Fawcett, 2006) and 2) False Negative Rate (FNR) (Powers, 2020).

In Figure 7, models such as GPT-4V, Claude3-Haiku, Yi-VL, and InternVL tend to answer "no" more frequently. Conversely, models like Emu2, MiniGPT-v2, and InstructBLIP are more inclined to answer "yes". Meanwhile, LLaVA-NeXT, CogVLM, Qwen-VL, and mPLUG-Owl exhibit a balanced performance without a strong bias towards either affirmative or negative classifications. Given that these models were not specifically trained for this task, the presence of such biases is not unexpected. This underscores the necessity of *MFC-Bench*, aiming to guide the enhancement of fact-checking capabilities in LVLMs for future developments.

### E.9 CASE STUDY

To better understand the reasoning process of the model in fact-checking, we are conducting a study on the correct and incorrect reasoning processes of the GPT-4V model. Figure 8 illustrates an instance where GPT-4V fails to identify manipulated content, specifically a face swap involving Joe Biden and another individual. This oversight underscores a significant limitation of GPT-4V in accurately recognizing individuals within images. The model's rationale primarily emphasizes overall scene consistency and plausible historical context, but it fails to detect the specific manipulation of Joe Biden's identity. In contrast, Figure 9 showcases GPT-4V's successful identification of manipulated content by accurately discerning the discrepancy between the emotional state depicted in the image and the corresponding caption. Todd Stern's smiling expression contrasts with the caption's

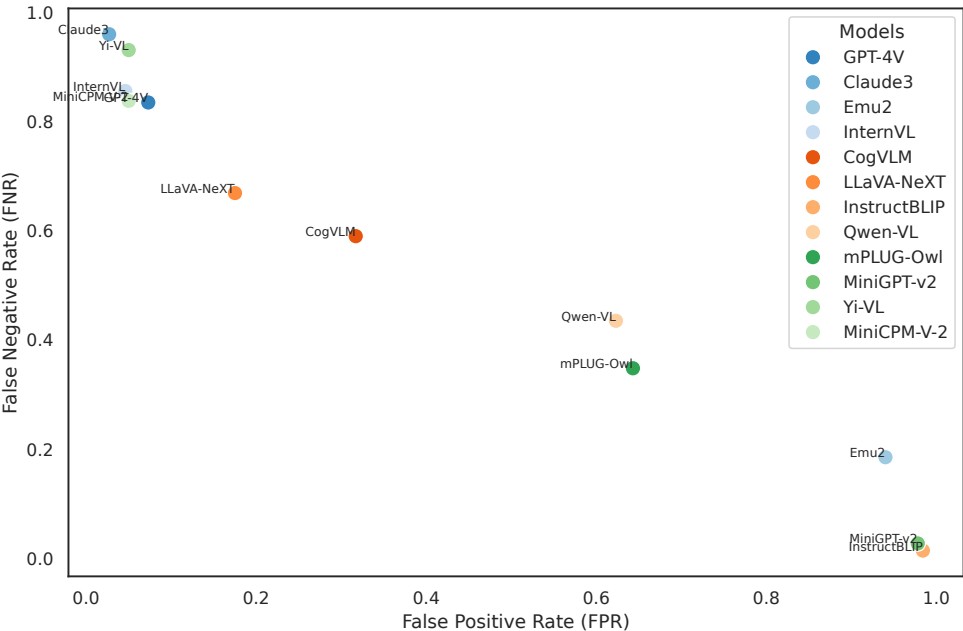

Figure 7: Yes/No Bias in tested LVLMs.

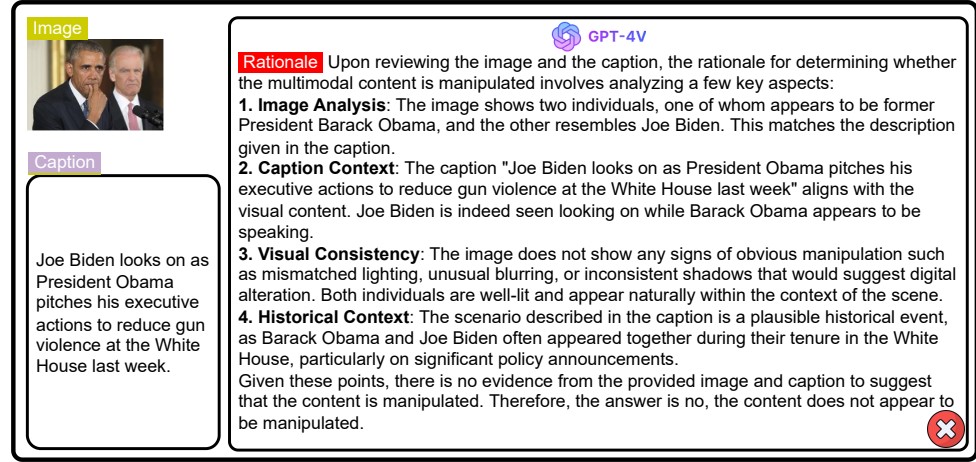

Figure 8: Example of wrongly predicted manipulation by GPT-4V with rationale.

description of him angrily rebuffing a suggestion. GPT-4V effectively recognizes this emotional mismatch, demonstrating its capability to evaluate the coherence between visual and textual elements.

### E.10   ERROR ANALYSIS

In zero-shot settings, the model's performance relies solely on its understanding of the instructions, its comprehension of the images and claims, and ultimately making a judgment based on this understanding(see also §E.9). The main results indicate that the model's fact-checking ability is weak. As discussed in §E.8, the Yes/No Bias also highlights this issue.

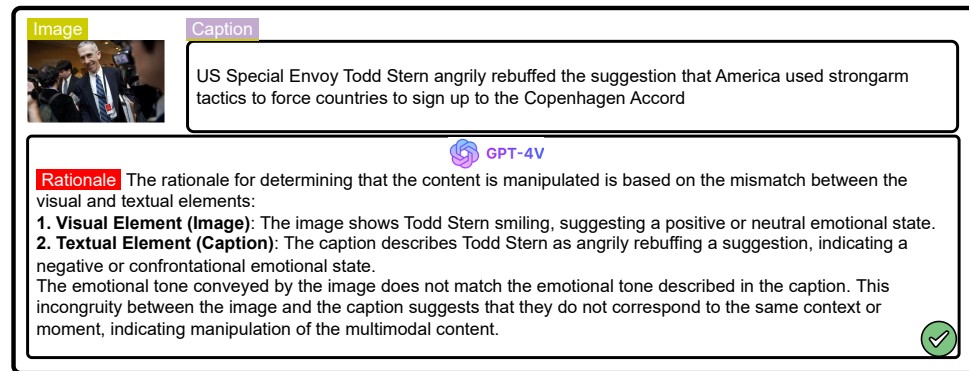

Figure 9: Example of correctly predicted manipulation by GPT-4V with rationale.

In few-shot settings, the model does not gain insights from the examples. For example, LLaVA-NeXT's usable response rate decreases, and it starts outputting gibberish instead of "yes" or "no". Specifically, in few-shot with CoT conditions, LLaVA-NeXT does not generate its own reasoning process but merely copies the rationale from previous examples. For example, one output from LLaVA-NeXT is, "Answer yes or no. Rationale: The image shows what seems to be an unnatural or edited blend of faces, particularly noticeable in the features of the man and the child. This indicates that the image may have been digitally altered.", which is already included in the demonstrations of the prompt.

## F  PROMPTS DESIGNED FOR MANIPULATION TECHNIQUES

1. Face Swap is a manipulation technique of cutting a face from one image and replacing it with a different face in another image. Your task is to determine if the claim and its image have used Face Swap. Answer yes or no.

2. Face Attribute Edit is a manipulation technique for altering facial expressions. Your task is to determine if the claim and its image have used Face Attribute Edit. Answer yes or no.

3. Background Change is a manipulation technique that involves altering the background of images. Your task is to determine if the claim and its image have used Background Change. Answer yes or no.

4. CLIP-based Stable Diffusion Generation is a manipulation technique that utilizes an image-to-image generation pipeline to produce manipulated images. Your task is to determine if the claim and its image have used CLIP-based Stable Diffusion Generate. Answer yes or no.

5. Textual Entity Replace is a manipulation technique that involves identifying named entities corresponding to persons in one text, locating these entities in another text, and swapping the surrounding contextual texts between the two. Your task is to determine if the claim and its image have used Textual Entity Replace. Answer yes or no.

6. Text Style Transfer is a manipulation technique that rewrites text to express the opposite sentiment. Your task is to determine if the claim and its image have used Text Style Transfer. Answer yes or no.

## G  DISCUSSION OF LABEL SETTING

We considered the following points in adopting this design philosophy for label setting:

- Simplicity and Clarity: As the first study to benchmark MFC with LVLMs, our design allows us to quantitatively assess the performance of LVLMs in a straightforward and intuitive manner. This simplicity facilitates preliminary in-depth analyses that more complex

settings might not easily provide. We find it exciting to cleverly and flexibly unify three significant data types under the MFC umbrella without adding unnecessary complexity.

- Poor Performance of LVLMs: Despite high F1 score of 84.8% on OOC Classification, the tasks are not too simple, as evidenced by lower F1 scores of 61.6% and 75.5% on Manipulation Classification and Veracity Classification. Besides, the best Accuracy and F1 on Manipulation Classification only achieve 64.0% and 56.6% by a lightweight LVLM, MiniCPM-V-2 (2.8B), leaving significant room to improve larger LVLMs that perform worse on this task.

- Appropriate Difficulty Levels: Our benchmark is designed to balance difficulty levels (i.e., OOC Classification: relatively easy; Veracity Classification: moderate; Manipulation Classification: relatively difficult), reflecting varying complexities to assess LVLM capabilities comprehensively. This integration allows for a broader evaluation of LVLMs' adaptability and generalization across diverse MFC data types.

- Foundation for Future Research: Our work lays the groundwork for future studies, which could incorporate more systematic human subject studies to explore interpretability and additional analytical dimensions. This potential for expansion underscores the value of our initial simplification and sets the stage for more complex investigations.

## H    DISCUSSION OF REAL-WORLD SCENARIOS

The main contribution of our benchmark is to provide insights into the trustworthy issue for current researchers studying existing emerging LVLMs. For a real-world fact-checking process, there are stages like claim detection, evidence retrieval, claim verification, justification production, etc. Our work just directly provides the check-worthy data so that the claim detection stage could be omitted. Then, the LVLM is evaluated by retrieving the inherent evidence embedded in its internal parameters, which can be regarded as the evidence retrieval stage in this benchmark work. Finally, for fact verification, the LVLM is used to verify the factuality in the verdict prediction stage with produced justification. Our human subjects evaluations have verified the soundness and alignment of the multimodal data for real-world needs.

## I    LIMITATIONS

As this is the first benchmark work to evaluate the multimodal fact-checking capacity of LVLMs, there are no doubt multiple efforts needed to improve the work in the future: 1) The dynamic and context-specific nature of multimodal fact-checking presents a challenge in interpretation and analysis. The current benchmark may not fully capture this complexity, potentially limiting the generalizability of our findings. Human interpretation of multimodal disinformation is inherently intricate and contextual. Real-world data from diverse domains will help advance this benchmark into various use case applications. Adding temporal dynamics will provide value when fact-checking historical facts. Additionally, future studies could be enhanced by a more comprehensive examination of bias and fairness in model evaluations to prevent the reinforcement or exacerbation of stereotypical hallucinations. 2) During the benchmarking process, we not only explore the three stages of verdict prediction for MFC: Manipulation Classification, OOC Classification, and Veracity Classification, but also investigate the last stage: Justification Production which requires the selected models to provide the post-hoc explanations. However, there might be a deeper model interpretability that is not touched in this work, which is to explain how an LVLM works internally. In future work, we should investigate the model's internal reasoning mechanisms and how it arrives at its conclusions from the perspective of the model architecture. Furthermore, the current LVLM demonstrates grounding capabilities that can be leveraged to better understand the model's interpretation of images and its fact-checking judgments. 3) Expanding the scope to include a broader array of models could enhance the robustness and applicability of the results. Incorporating diverse multilingual datasets, the audio modality, and emerging LVLMs into our benchmark work could provide a more nuanced understanding of LVLMs' capabilities across various languages. Although there is a long way to go, where there is a will, there is a way.

