# OpenReview forum: "MFC-Bench: Benchmarking Multimodal Fact-Checking with Large Vision-Language Models"
_ICLR.cc/2025/Workshop/BuildingTrust — BuildingTrust_

### Official Review · Reviewer_ftPU · 2025-02-27
**Introduces a new vision-language benchmark to test for VLMs' ability to detect input manipulations/misalignments.**

**Rating:** 6
**Confidence:** 4

**Review:**

Introduces a new benchmark. The task may not have much ecological validity, and hence hard to understand the importance of it, but it is maybe useful to test for model robustness to certain input manipulations. The human eval for assuring data quality is well done. I also like the evaluation of the justification across the different dimensions. Good thorough analysis.

---

### Decision · Program_Chairs · 2025-03-04

**Decision:**

Accept

**Comment:**

MFC-Bench provides a useful tool for evaluating model trustworthiness, particularly in detecting factual inconsistencies in multimodal settings. Further discussion on the real-world applicability of the benchmark would enhance its contribution. The paper is highly relevant for the workshop.